# UniT: A Unified Look at Certified Robust Training against Text Adversarial Perturbation

**Muchao Ye**[1]    **Ziyi Yin**[1]    **Tianrong Zhang**[1]    **Tianyu Du**[2]
**Jinghui Chen**[1]    **Ting Wang**[3]    **Fenglong Ma**[1]*
[1]The Pennsylvania State University, [2]Zhejiang University, [3]Stony Brook University
{muchao, zmy5171, tbz5156, jzc5917, fenglong}@psu.edu,
zjradty@zju.edu.cn, twang@cs.stonybrook.edu

## Abstract

Recent years have witnessed a surge of certified robust training pipelines against text adversarial perturbation constructed by synonym substitutions. Given a base model, existing pipelines provide prediction certificates either in the discrete word space or the continuous latent space. However, they are isolated from each other with a structural gap. We observe that existing training frameworks need unification to provide stronger certified robustness. Additionally, they mainly focus on building the certification process but neglect to improve the robustness of the base model. To mitigate the aforementioned limitations, we propose a unified framework named UniT that enables us to train flexibly in either fashion by working in the word embedding space. It can provide a stronger robustness guarantee obtained directly from the word embedding space without extra modules. In addition, we introduce the decoupled regularization (DR) loss to improve the robustness of the base model, which includes two separate robustness regularization terms for the feature extraction and classifier modules. Experimental results on widely used text classification datasets further demonstrate the effectiveness of the designed unified framework and the proposed DR loss for improving the certified robust accuracy.[†]

## 1 Introduction

Despite the tremendous performance of deep neural networks (DNNs) in natural language processing (NLP) tasks, their robustness has been doubted due to their vulnerability against adversarial attacks [15]. Particularly, a type of word-level adversarial perturbation named *synonym substitution* can generate adversarial examples with high semantic similarity even in the hard-label setting with limited information [14, 26]. Accordingly, recent years have seen an urge for robust NLP models that can provide certified robust predictions [23, 25, 31] for this type of attack. A prediction for a text sample is **certified** if the hard-label prediction is correct and remains unchanged when the input is changed to any text sample constructed from the synonym substitution. The key to producing certified predictions is *certified robust training*, which introduces perturbation during training to ask the model to adapt to it and enables it to still perform well in the inference stage under perturbation.

Due to the large scale of NLP models nowadays, certification methods adopted for text data are usually probabilistic ones. In this context, given a base model that outputs a single prediction, a smoothed model is built on it in the inference stage with randomized mechanisms, whose outputs are used to decide whether predictions are certified. Thus, two aspects are essential for generating certified predictions: the robustness of the base model and the certification of the smoothed model.

---

*Corresponding author.

[†]The implementation code is available at https://github.com/machinelearning4health/UniT.

37th Conference on Neural Information Processing Systems (NeurIPS 2023).

Existing studies have established the foundation for designing smoothing techniques and their corresponding certification conditions. As Figure 1(a) manifests, a base model usually consists of three modules: (1) an embedding module that converts discrete words into continuous embeddings, (2) a feature extraction module that obtains its deep representation, and (3) a classifier constructed by the fully connected (FC) layer that converts the deep representation to logits. Given that, randomized mechanisms can either build in the discrete word space (**Type I**) or the continuous latent space (**Type II**). Figure 1(b) shows that Type I pipelines, including SAFER [25] and WordDP [23], use *perturbed text samples* generated by replacing original words with their corresponding synonyms to train the base model and construct a smoothed model for certification. Another type of training, the Type II one, is shown in Figure 1(c), which are recently proposed by CISS [31]. They *add Gaussian noise to latent features of original text samples* to train the base model and certify the prediction in the latent space in a randomized smoothing [1] fashion with the help of the interval bound propagation (IBP) [5] module.

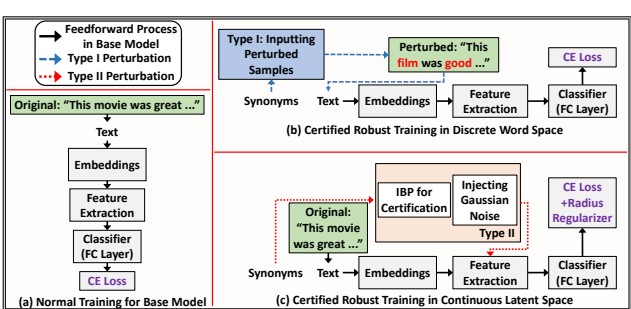

Figure 1: Given the (a) base model, (b) Type I frameworks construct the smoothed model in the discrete word space while (c) Type II frameworks construct it in the latent space with IBP. There is a need for unifying these training frameworks and improving the robustness of the base model.

Given these profound pioneering works, the observation motivating our work is that existing pipelines *ignore the unification of training frameworks and robustness of the base model*. For one thing, structure-wise speaking, Type II frameworks like CISS need to include an extra IBP module compared to Type I frameworks, which affects the certification because of the loose bound problem [30] of IBP. This is validated by our finding that IBP can cause the failure to strike a balance between natural accuracy and certification ratio. For another, both types of frameworks mainly use the cross-entropy (CE) loss calculated from the perturbed inputs for training to improve the discriminative ability of the base model, which is the same as normal training. However, this strategy lacks fine-grained robustness regularization for individual modules and consideration of the final certification target.

To develop a new method that mitigates the aforementioned limitations, we will face two research questions: (**RQ1**) how to build a unified framework for these two types of pipelines to provide stronger certified robustness and (**RQ2**) how to design robustness regularization terms for individual modules to further improve the base model robustness. Our solution to them is as follows:

For **RQ1**, we propose a new framework named UniT to unify Type I and Type II training. UniT resorts to the embedding space as an intermediate for unification. In the Type I scenario, it uses embeddings of perturbed words to train the base model directly. For the Type II scenario, it does not need the IBP module and adds Gaussian noise directly to the concatenated word embeddings of the text sample. We accordingly provide a novel certification condition in the embedding space.

For **RQ2**, we design a novel loss for certified robust training named decoupled regularization (DR) loss that compensates the CE loss with the modular regularization (MR) term. It consists of two parts. Part I is the $l_2$ norm between the representations of the original sample and the perturbed input, which concentrates on the feature extraction module. Part II includes a novel term calculating the prediction margin from the logits when the original sample representation is slightly perturbed (modeled by Gaussian noise), which takes the certification target into consideration and aims to regularize the classifier module. They are linearly combined to provide supportive regularization for different modules based on their responsibilities and refine the robustness. To sum up, our contributions are:

- To the best of our knowledge, we are the first to provide a comprehensive viewpoint on certified robust training in text. We design a unified structure named UniT built in the embedding space that enables Type II frameworks to be conducted without using IBP during training and certification.

- We propose the DR loss to improve the robustness of the base model. It provides modular regularization directly to aid the CE loss. That is, the feature extraction module is regularized by

the representational $l_2$ norm, while the classifier module is regularized based on the prediction margin calculated from the perturbed representations obtained by adding noise to the original ones.

- Experimental results show that the design of the UniT with DR loss is effective in improving the certified robust accuracy of both types of certification scenarios.

## 2    Preliminaries and Related Work

### 2.1    Certified Robust Training in Image

The development of certified robust training dates back to certifying multi-layer perceptions (MLPs) for image data. Early work tries to formulate certification as an optimization problem with the techniques of mixed integer linear programming [21] and semidefinite relaxation [18]. A practical solution to this problem is using the IBP [3] method and its variants [8], whose key idea is to regard the upper and lower bounds of activated output in each layer as parameters and they are trained by backpropagation. Later development on deterministic certification methods also includes developing Lipschitz networks [10]. However, they can only certify the robustness of shallow neural networks. Another line of certified robust training is conducted by randomized smoothing [1]. It helps provide a certificate in a probabilistic manner, which is more scalable and can certify larger DNNs.

### 2.2    Certified Robust Training in Text

**Problem Formulation.** Suppose that we are given a text classification dataset $\mathcal{D}$. Let $\mathcal{Y} = \{y_1, \cdots, y_c\}$ denote the set of $c$ classes of $\mathcal{D}$. We have $(X, y)$ as a sample drawn from $\mathcal{D}$. For the text sample $X = [w_1, \cdots, w_n]$ with $n$ words, $w_i$ is the $i$-th word, and its ground truth is $y \in \mathcal{Y}$. We denote $f$ as the **base model** for the text classification task, which outputs the prediction logits $[f_{y_1}(X), \cdots, f_{y_c}(X)]$ for each class in $\mathcal{Y}$. Suppose that $f$ can make the correct prediction, i.e., $\arg\max_{y_i \in \mathcal{Y}} f_{y_i}(X) = y$. In the context of **certified robustness**, we are interested in getting a certified prediction result such that $\arg\max_{y_i \in \mathcal{Y}} f_{y_i}(X') = y$ holds for any allowed perturbed sample $X'$ of $X$. In this paper, we center on certifying perturbed samples that are constructed by a powerful adversarial attack named synonym substitutions [9] (an example is shown in Table 10 in the Appendix). Note that this setting is different from the one in [27] which certifies the prediction under unconstrained word perturbation for a limited amount of words, and adversarial examples generated from synonym substitution have lower perturbation rates compared to the ones from rephrasing [17] and prompting. In this setting, for each word $w_i \in X$, suppose it has $m_i$ synonyms, its allowable adversarial substitution set is denoted by $S(w_i) = \{w_i^{(1)}, \cdots, w_i^{(m_i)}\}$ (Note that $S(w_i)$ includes $w_i$). All perturbed samples $X'$ obtained by replacing original words with their synonyms construct the allowed perturbed sample set $A(X) = \{X'|X' = [w_1', \cdots, w_n'], w_i' \in S(w_i), i = 1, \cdots, n\}$.

**Training and Certification.** Since $A(X)$ is a large set of perturbed samples ($|A(X)| = \prod_{i=1}^{n} m_i$), it is impractical to certify the prediction by asking $f$ to make a prediction on every $X' \in A(X)$. Therefore, existing techniques usually certify prediction results by the upper bound condition. If such a certification condition satisfies, the prediction is certified robust. Certified robust training is needed to attain a certified prediction result for $X$ such that the certification condition can be satisfied in testing. For example, [4] and [5] introduce the IBP-based methods and use the upper bound of text representation in the final layer to certify the result. However, due to the costly bound propagation, IBP-based methods can only be used in shallow MLPs and cannot be applied to large language models like BERT [2]. Besides, IBP has a loose bound problem [30], i.e., upper bounds get loose in the last several layers during propagation, which obstructs the certification of some samples.

Thus, recent studies mainly use probabilistic certified defense methods to obtain certified predictions for more powerful but intricate networks. As demonstrated in Figure 1, existing certified robust training methods for probabilistic certification are categorized into two types. Their common treatment is to introduce perturbation during training. They ask the whole network to get used to the perturbed samples, which are either perturbed input texts $X'$ (**Type I**) [23, 25] or perturbed latent features injected into Gaussian noise (**Type II**) [31]. They usually use the CE loss to improve the discriminative ability of the base model. In certification, they construct a **smoothed model** $F$ for $f$. *A prediction is certified when the logit score of the ground truth class output by $F$ is larger than that of the runner-up class by a large enough margin*, which we refer to as the certification condition. Denote $F_{y_i}$ as the soft-label prediction score of $F$ on class $y_i$. The runner-up class is defined as

$y_B = \arg\max_{y_i \in \mathcal{Y}, y_i \neq y} F_{y_i}(X)$ with the largest logit except for $F_y$. We now briefly mention their certification and kindly ask readers to review their works for a more complete picture if needed.

(1) For the Type I scenario, $F$ is constructed by averaging the results attained from inputting several perturbed text samples to $f$. The certification *depends explicitly on the prediction margin* $F_y(X) - F_{y_B}(X)$. For example, the certification condition of SAFER (Proposition 1 in [25]) is $F_y(X) - F_{y_B}(X) > 2q_X$, where $q_X$ is a constant calculated from the synonym set of the input $X$.

(2) In the Type II scenario, $F$ is constructed by repeatedly adding Gaussian noise $\delta \sim \mathcal{N}(0, \sigma^2 I)$ to the text representation in the latent space, where $\sigma$ is the standard deviation and $I$ is the identity matrix. To illustrate, under the randomized smoothing framework, CISS [31] repeatedly draws Gaussian noise perturbations and adds them to the latent feature of the original sample. It counts the top-1 prediction distribution of $\mathcal{Y}$ given all perturbed inputs. After that, the prediction is certified if $\underline{p_y} > 0.5$ and $R = \sigma \Phi^{-1}(\underline{p_y}) > \hat{R}$ (Algorithm 2 in [31]), where $\underline{p_y}$ is the lower bound of the expected value of the event that the hard-label prediction output by $f$ of the given perturbed input is class $y$, $R$ is the certified radius, $\Phi^{-1}$ is the inverse of the standard Gaussian cumulative distribution function (CDF), and $\hat{R}$ is the maximum $l_2$ norm between $X$ and any $X' \in A(X)$ in the latent space obtained from IBP. The certification *implicitly depends on the prediction margin*. If the prediction margin is large, the calculated $\underline{p_y}$ and $R$ will get greater, which helps the certification.

There are also some interesting discussions extended from certified robustness such as whether certified robustness helps fairness [16]. We mainly focus on the certification performance herein and kindly ask readers to review related works for further information about those discussions.

## 3  UniT: Unified Certified Robust Training

To provide stronger certified robustness for those important but independent frameworks, we design a unified one named UniT that can unify these two types of training without introducing extra modules such as the IBP module used in CISS. In this paper, we utilize BERT as the base model because it yields the best performance for state-of-the-art training frameworks such as SAFER and CISS in two settings. The basic processing procedure of the base model is shown in Figure 2, indicated by the black solid lines. It first transforms the input sample with

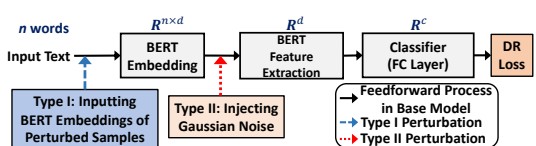

Figure 2: Given the base model, UniT unifies two frameworks by working in the embedding space. Type I training replaces original embeddings with embeddings of perturbed samples. Type II training adds Gaussian noise to original word embeddings.

$n$ words to a concatenated BERT word embedding $x \in \mathbf{R}^{n \times d}$ where $d = 768$ is the dimension of word embedding. Then the word embeddings will be processed by the BERT feature extraction module (Transformer [22, 24] blocks) which outputs the [*CLS*] token embedding $z \in \mathbf{R}^d$ as the representation of the whole text. Finally, the classifier module (FC layer) outputs logits from $z$, and the loss is calculated for training. We shall notice that the structural decomposition analysis above is general among different models, e.g., RoBERTa [11] and ALBERT [7], so we can also adopt DR loss for other structures when needed. We now detail our motivation and design of UniT.

### 3.1  Motivation: Why Do We Need Unification and Better Supervision?

The supervision for improving the discriminative ability of the model during certified robust training generally builds on the CE loss. Since the calculation of the loss function is only related to the last FC layer in the classifier, we denote the last FC layer as $g$ for illustration. We denote $z' \in \mathbf{R}^d$ as the representation obtained from the feature extraction module given a perturbed input.

The CE loss is calculated from $z'$, i.e., $\ell_{ce} = -\log(\frac{\exp(g_y(z'))}{\sum_i \exp(g_{y_i}(z'))})$, where $[g_{y_1}, \cdots, g_{y_c}]$ is the logit distribution given the perturbed input for each class in $y_i \in \mathcal{Y}$. As depicted in Figure 1, Type I training builds a smoothed model in the discrete word space, and its training is almost the same as normal training except for perturbed inputs. Meanwhile, the loss for Type II training is $\ell_{tr} = \ell_{ce} + \gamma \cdot \ell_{\hat{R}}$, where $\gamma$ is the weight hyperparameter. It contains not only the CE loss but also a certified radius regularizer $\ell_{\hat{R}} = \max(\hat{R} - R + \mu, 0)$, where $\mu$ is the relaxation hyperparameter, $R$ is the certified

radius, and $\hat{R}$ is the maximum deviation caused by synonym substitution in the latent space that we introduce in Sec. 2.2. We notice that there is a trade-off between the certification ratio and the natural accuracy for $\ell_{tr}$ in Type II training: *a relaxed regularization on the certified radius can make the base model have higher natural accuracy but a lower certification ratio, and vice versa.*

We verify this observation by taking CISS for instance. We can obtain a relaxed regularization on certified radius by replacing $\hat{R}$ with its **lower bound** $\hat{R}_{lb}$ and a strict one by its **upper bound** $\hat{R}_{ub}$. Suppose the minimum and the maximum deviation in any dimension is $\underline{\rho}$ and $\bar{\rho}$, respectively, and the number of dimensions in the latent space is $d_I$. We then have $\hat{R}_{lb} = \sqrt{d_I} \cdot \underline{\rho}$ and $\hat{R}_{ub} = \sqrt{d_I} \cdot \bar{\rho}$. We denote the corresponding regularizers as $\ell_{\hat{R}_{lb}} = \max(\hat{R}_{lb} - R + \mu, 0)$ and $\ell_{\hat{R}_{ub}} = \max(\hat{R}_{ub} - R + \mu, 0)$.

| Regularization | $\ell_{\hat{R}}$ | $\ell_{\hat{R}_{ub}}$ | $\ell_{\hat{R}_{lb}}$ |
|---|---|---|---|
| Natural Acc. (%) | 93.88 | 89.57 | **96.34** |
| Cert. Ratio (%) | 94.38 | **99.56** | 0.81 |
| CRA (%) | 88.60 | **89.18** | 0.78 |

Table 1: Trade-off between natural accuracy and certification ratio. Experiments are conducted in Yelp [19] with CISS.

We show the results of how this trade-off affects certified robust accuracy in Table 1. Compared to the original $\ell_{\hat{R}}$, if we impose a relaxed regularization $\ell_{\hat{R}_{lb}}$, the natural accuracy will increase but the certification condition is hard to satisfy, and consequently, the certified robust accuracy is near $0\%$. On the contrary, a stricter regularization $\ell_{\hat{R}_{ub}}$ makes the smoothed model satisfy the certification condition easily (almost 100%), but it hurts the natural accuracy. An interesting finding in Table 1 is that strict regularization $\ell_{\hat{R}_{ub}}$ helps obtain a higher certified robust accuracy (CRA) than the original regularization $\ell_{\hat{R}}$ used in CISS. This is because $\hat{R}$ in CISS is obtained from IBP, and it has a loose bound problem. The certified radius regularization $\ell_{\hat{R}}$ from IBP is a somewhat strict one, so CISS cannot strike a balance for this trade-off with IBP. Therefore, **we shall remove IBP and avoid this calculation**. In addition, since the CE loss is deployed to improve discriminative ability, another direction for mitigation is to adopt better supervision other than the CE loss for ameliorating the model robustness, which motivates us to **provide individual supervision for each module**.

### 3.2 Training Input

Recall that in the **Type I** scenario, the original input $X = [w_1, \cdots, w_n]$ is replaced by the perturbed sample $X' = [w'_1, \cdots, w'_n] \in A(X)$. Correspondingly, in this scenario, UniT uses the embedding $x' = [u(w'_1), \cdots, u(w'_n)]$ as the input to train the base model, where $u$ is the *BERT embedding* function. For the **Type II** scenario, to facilitate randomized smoothing, UniT needs to inject Gaussian noise into the *BERT embeddings* for training. Thus, given a sample of zero-mean Gaussian noise $\delta \sim \mathcal{N}(0, \sigma^2 I) \in \mathbf{R}^{n \times d}$, UniT uses the embedding $x + \delta$ as the input. In this sense, we unify two types of certified robust training in one framework by utilizing the embedding space as the intermediate. Thus, all modules can remain the same for either type of certified robust training.

### 3.3 Certification

For the **Type I** scenario, we can obtain the certified robustness guarantee from Proposition 1 of SAFER [25] by constructing the smoothed model based on synonym substitutions. We thereby intend to detail our proposed certification for the **Type II** scenario without IBP. Note that the key calculation for certification in this scenario is $\hat{R}$, which is the maximum $l_2$ norm caused by allowed perturbed synonyms in the embedding space. This calculation by UniT is different from CISS because UniT does not need IBP module, and it is directly obtained in the BERT embedding space $\mathbf{R}^{n \times d}$ as follows.

Given the text sample $X = [w_1, \cdots, w_n]$, its BERT embedding is $x = [u(w_1), \cdots, u(w_n)]$. Similarly, for any perturbed sample $X' = [w'_1, \cdots, w'_n] \in A(X)$, its embedding $x' = [u(w'_1), \cdots, u(w'_n)]$. Since each word is independent of the others in the embedding space, the $l_2$ norm between $x$ and any $x'$ is $||x' - x|| = \sqrt{\sum_{i=1}^{n} ||u(w_i) - u(w'_i)||^2}$. For each word $w_i \in X$, because its synonym embedding set is $U_i = \{u(w_i^{(1)}), \cdots, u(w_i^{(m_i)})\}$, the maximum deviation between $w_i$ and any word in $S(w_i)$ measured by $u$ is $||v_i|| = \max_{e \in U_i} ||u(w_i) - e||$. Due to the independence of embedding of each word, the maximum deviation caused by all synonyms is

$$\hat{R} = \sqrt{||v_1||^2 + \cdots + ||v_n||^2}. \tag{1}$$

By adopting Eq. (1) to calculate $\hat{R}$, we can train and certify in the Type II scenario without IBP and avoid bound propagation. We can conduct the certification in the embedding space with Theorem 1.

**Theorem 1** *Given $(X, y)$ from $\mathcal{D}$, suppose that the text embedding of $X$ is $x$. Let $h : s \to \mathcal{Y}$ be the hard-label prediction function of UniT in the Type II scenario that takes as the input a vector $s \in \mathbf{R}^{n \times d}$ in the text embedding space and outputs the class with the largest logit. Let $\delta \sim \mathcal{N}(0, \sigma^2 I)$, where $\sigma$ is the standard deviation of the added Gaussian noise. If*

$$\mathbb{E}_\delta[h(x + \delta) = y] \geq \max_{y_i \in \mathcal{Y}, y_i \neq y} \mathbb{E}_\delta[h(x + \delta) = y_i], \tag{2}$$

*then $\forall x', h(x') = y$ is robust where $x'$ is the text embedding of any $X' \in A(X)$ if $\hat{R} \leq R$, and*

$$R = \frac{\sigma}{2}[\Phi^{-1}(\mathbb{E}_\delta[h(x + \delta) = y]) - \Phi^{-1}(\max_{y_i \in \mathcal{Y}, y_i \neq y} \mathbb{E}_\delta[h(x + \delta) = y_i])], \tag{3}$$

*where $\Phi^{-1}$ is the inverse of the standard Gaussian CDF, and $\hat{R}$ is calculated from Eq. (1). Due to the space limit, the proof and certification algorithm are shown in Sec. A.7 in the Appendix.*

### 3.4 Training Loss

To provide better supervision for training the base model, we propose a decoupled regularization (DR) learning paradigm that directly conducts modular regularization to aid the CE loss. We decouple the regularization on the whole model into one on the feature extraction module and the other on the classifier module. Thus, this pathway includes two branches. Such a regularization decomposition requires each module to be robust under the perturbation, which contributes to enhancing the robustness of the base model with more flexibility, leading to better training and certification results.

**Feature Extraction Regularization.** For the feature extraction module, its responsibility is to output a deep representation of a given text. The stability of representation, i.e., the degree of change in the representation space given a small change of input, depicts the robustness of the feature extraction module. To regularize it, we calculate the $l_2$ norm between perturbed representation $z'$ and original representation $z$,

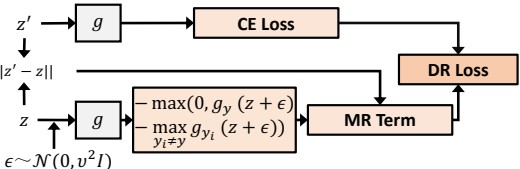

Figure 3: DR loss contains a pathway (MR term) for providing modular regularization for CE loss.

$$\ell_{fr} = ||z' - z||, \tag{4}$$

where $|| \cdot ||$ is the $l_2$ norm.

We choose $l_2$ norm over cosine similarity because it is directly related to the prediction margin defined in Sec. 2.2 in calculation. Thus, it can limit the influence of perturbation on downgrading the certification performance. Intuitively, it functions as a regularization term that asks the representations obtained from the feature extraction module given the original sample and perturbed sample to be similar to each other, which is helpful for obtaining a similar logit for $z'$ given the same classifier module. Hence, it further enhances the representational stability and the model robustness.

**Classifier Robustness Regularization.** Since the representational stability regularization in Eq. (4) only concerns the feature extraction module, another direction for strengthening the base model robustness is to design a feedback signal in the classifier level that is complementary to Eq. (4). We know that the classifier module is the one that outputs the logits. Recall that in Sec. 2.2 we discuss that the certification is highly dependent on the prediction margin output by the smoothed model, which is the difference between the logits of the ground truth class and the runner-up class. By connecting these two observations together, in the context of certified robust training, we can interpret the robustness of the classifier module as being able to generate a positive prediction margin despite the perturbation added to the original input $z$.

Consequently, we introduce a novel regularization term for the classifier module based on its ability to output a positive prediction margin against a small perturbation. Since this regularizer is confined to the classifier, we introduce a small amount of Gaussian noise $\epsilon$ here to model the perturbation and sample a point $z + \epsilon$ from the neighbor of $z$. *Note that $\epsilon$ only serves for modeling small*

*perturbation in training. It will not be added in certification and is not related to the Gaussian noise $\delta$ used for certification in Type II scenario.* That is, it starts with adding Gaussian noise $\epsilon \in \mathbf{R}^d$ to $z$, where $\epsilon \sim \mathcal{N}(0, \nu^2 I)$ is drawn from the zero-mean isotropic Gaussian distribution with $\nu$ as the standard deviation hyperparameter. Denote the prediction margin given a vector $k \in \mathbf{R}^d$ as $\mathcal{M}(k) = g_y(k) - \max_{y_i \in \mathcal{Y}, y_i \neq y} g_{y_i}(k)$, where $g_y(\cdot)$ and $\max_{y_i \in \mathcal{Y}, y_i \neq y} g_{y_i}(\cdot)$ refers to the logits of the ground truth class and the runner-up class, respectively. We then calculate the margin given perturbed input $z + \epsilon$ as $\mathcal{M}(z + \epsilon)$. We take its negative as an optimization term for minimization:

$$\ell_{cr} = -\max(\mathcal{M}(z + \epsilon), 0). \tag{5}$$

$\ell_{cr}$ has a $\max$ operation that outputs 0 when $\mathcal{M}(z + \epsilon) < 0$. This operation acts as feedback for asking the optimization to concentrate on improving the representation space and the discriminative ability through the other terms when $\mathcal{M}(z + \epsilon) < 0$, which will not regularize $g$ if $z$ is not well learned.

**Modular Regularization Term.** After that, these two regularizers are combined as

$$\ell_{mr} = \max(\ell_{fr} + \alpha \cdot \ell_{cr}, \xi), \tag{6}$$

which we refer to as the modular regularization (MR) term. In Eq. (6), the MR term is calculated from the linear combination of $\ell_{fr}$ and $\ell_{cr}$, where $\alpha$ is a weight hyperparameter greater than 0. Additionally, we introduce relaxation during optimization with $\max$ operation, and $\xi > 0$ is the relaxation hyperparameter. Putting them together provides a regularization on how small $||z' - z||$ should be given the margin information. This geometric interpretation is detailed in Sec. A.8.

**DR loss.** Given the MR term, we can combine it with the CE loss as the DR loss, which is

$$\ell_{dr} = \ell_{ce} + \beta \cdot \ell_{mr}, \tag{7}$$

where $\beta$ is a weight hyperparameter for $\ell_{mr}$ term. Given that, in the Type I training, we can directly use $\ell_{dr}$ for training. For Type II training, since the certification is related to $\hat{R}$, we also have a regularizer for $\hat{R}$ with $\ell_{\hat{R}} = \max(\hat{R} - R + \mu, 0)$ as CISS does, where $R$ is calculated from Eq. (3) and $\mu$ is the relaxation hyperparameter. Thus, the training loss in the Type II scenario is $\ell_{tr} = \ell_{dr} + \gamma \cdot \ell_{\hat{R}}$, where $\beta$ and $\gamma$ are hyperparameters.

**Remark.** From Eq. (7), we observe that the DR loss is composed of the CE loss and the MR term, which are indispensable to each other. If we only use the CE loss, the individual regularization will be ignored. With the help of the MR term, the introduced regularizers in Eq. (6) directly request the representation to be stable and the classifier to be robust under the perturbation, which helps the CE loss to improve the base model robustness. Meanwhile, since the MR term does not penalize the wrongly classified cases, the CE loss is necessary for improving the discriminative ability.

As for computation cost, we acknowledge that compared to CE loss calculating the proposed DR loss needs to increase the computation complexity due to the calculation of getting the features of perturbed input and original input during training. Such a tradeoff is natural: to obtain better robustness it is generally unavoidable to increase computation cost, and the increase of computation cost also occurs for existing training techniques for improving robustness such as using PGD [13] loss and TRADES [28] loss. We want to point out that the computation cost of calculating DR loss is relatively lower which is approximately twice of the cost of caluating CE loss, which is similar to that of single-step PGD and is acceptable in implementation.

## 4 Experiments

Given the proposed method above, the questions of interest include: **(Q1)** Does the unified framework UniT outperform existing state-of-the-art approaches in both scenarios? **(Q2)** How does UniT improve the robustness of the base model in certified robust training? **(Q3)** Is the design of the UniT reasonable? We now answer them with the following experimental results.

### 4.1 Experimental Setup

We describe the experimental setup first. Due to the space limit, we kindly ask the readers to refer to the Appendix for the details of hyperparameters, dataset description, and metric calculation.

**Datasets.** We conduct experiments on four widely used text classification datasets: (1) IMDB [12], (2) SST2 [20], (3) Yelp [19] and (4) AG [29].

**Baselines.** Because of different certification conditions used in different works, to gain a fair comparison, we compare the proposed method with baselines individually in Type I and Type II scenarios. To illustrate: (1) Since the code of WordDP [23] is unavailable, the Type I baseline we use is SAFER [25], which is the ground-breaking method that trains a model by inputting perturbed text samples constructed by randomly sampled synonyms and certifies the prediction results directly by the prediction margin. (2) The Type II baseline is CISS [31], a recently proposed method that trains the model and certifies the prediction with IBP in a randomized smoothing manner. Their implemented base models are BERT because it yields the best performance. Since [31] has shown that IBP methods [4, 5] cannot certify large networks like BERT, we hereby do not discuss them. We also include adversarial training losses like PGD loss [13] and TRADES loss [28] as baselines.

**Metric.** Naturally, certified robust accuracy (CRA) is the used comparison metric, and *certified robust accuracy = natural accuracy × certification ratio*. It denotes the fraction of predictions that are not only correctly classified but also certified robust. As for its calculation, we follow the same process used in the baseline methods to have a fair comparison. Further details are in Appendix.

### 4.2 Comparison Results

**Comparison of Certified Robustness in Type I Scenario.** To answer Q1, we first conduct the comparison experiments in Type I scenario. For SAFER and UniT, both of them use BERT as the base model, and the difference is that SAFER is trained with the CE loss and UniT is with the DR loss. The comparison results are in Table 2, which show that the DR loss increases the certified robust accuracy up to 3.68%, 1.37%, 0.68%, and 0.53% in IMDB, SST2, Yelp, and AG datasets, respectively. These results indicate that the MR term enlarges the prediction margin from the CE loss under perturbation to yield a higher certified robust accuracy. Hence, the DR loss can improve the robustness of the base model and help UniT achieve better performance in Type I scenario.

We also include adversarial training losses as baselines for Type I Scenario. From Table 2, DR loss can consistently perform better than PGD loss and TRADES loss in four datasets. Especially, DR loss is able to obtain a relatively high increment in terms of CRA on the datasets with long text samples such as IMDB and SST2. Thus, DR loss is a more suitable choice for certified robust training for its decomposed regularization, which requires each module to be robust under perturbation.

| Base Model | Loss | IMDB | SST2 | Yelp | AG |
|---|---|---|---|---|---|
| | CE Loss (SAFER) | 85.36 | 91.65 | 97.19 | 93.78 |
| BERT | PGD Loss | 87.52 | 90.28 | 97.86 | 93.98 |
| | TRADES Loss | 86.80 | 90.44 | 97.56 | 93.96 |
| | DR Loss (UniT) | **89.04** | **93.02** | **97.87** | **94.31** |

Table 2: Comparison of certified robust accuracy (%) in the Type I scenario.

**Comparison of Certified Robustness in Type II Scenario.**
We further conduct the comparison experiments in the Type II scenario to answer Q1, and the results are shown in Table 3. During implementation, we find that the training for randomized smoothing in CISS can only work in relatively large datasets (100k+ training samples). Therefore, we can only report comparison results on two large datasets satisfying this condition, i.e., Yelp and AG, with CISS as the baseline. As we have seen before, directly replacing the CE loss (first row) with the DR loss (second row) can help improve CRA.

| Method | Loss | Yelp | AG |
|---|---|---|---|
| CISS | CE | 88.60 | 82.47 |
| CISS | DR | 89.22 | 82.93 |
| UniT | DR | **91.24** | **84.32** |

Table 3: Comparison of certified robust accuracy (%) in the Type II scenario.

In this setting, we are more interested in the analysis of whether using the structure of UniT can make Type II training achieve stronger certification without using extra modules. By comparing the last two rows in Table 3, we find that UniT achieves an increase of 2.02% and 1.39% in Yelp and AG in terms of certified robust accuracy than the CISS structure trained with the DR loss, respectively. This improvement shows that by directly conducting randomized smoothing in the BERT embedding space, UniT not only adapts to both Type I and Type II training but also improves the certified robust accuracy by avoiding the loose bound problem of the IBP module.

**Comparison of Empirical Robustness.** We also conduct experiments on comparing empirical robustness. We employ BERT as the base model and use PGD loss and TRADES loss as baselines. In our experiments, we conduct adversarial attacks by utilizing TextBugger [9], a commonly used method for generating text adversarial examples. Following the existing adversarial attack setup, we randomly draw 1,000 test samples for adversarial example generation. The defense results of different methods are shown in Table 4, which demonstrate the empirical robust accuracy of DR loss is higher than those of PGD loss and TRADES loss. Compared to baselines, the improvement is obtained by the individual robustness supervision for each module provided by DR loss.

| Base Model | Loss | IMDB | SST2 | Yelp | AG |
|---|---|---|---|---|---|
| | PGD Loss | 62.0 (87.1) | 87.6 (91.5) | 92.4 (96.9) | 85.5 (93.7) |
| BERT | TRADES Loss | 57.8 (84.8) | 86.3 (91.7) | 91.9 (97.0) | 84.4 (94.3) |
| | DR Loss | **72.7** (86.9) | **89.8** (93.3) | **96.9** (98.4) | **87.6** (92.9) |

Table 4: Comparison of empirical robust accuracy (%) with adversarial training losses. We also show the corresponding natural accuracy indicated by the parentheses.

### 4.3 Insight Analysis on UniT

**Analysis on Certification Condition.** For Q2, we first find that the improvement is achieved by improving the certification ratio in the Type II scenario. As shown in Theorem 1, UniT includes a stronger certification condition in the Type II scenario that does not require the calculation from IBP. We now verify it by comparing the certification ratio between CISS and UniT. The certification ratio results shown in Table 5 indicate that UniT has a larger ratio of correctly classified test samples that increases up to 2.84% and 4.23% compared to CISS, and 2.06% and 3.85% compared to CISS trained with

| Method | Loss | Yelp | AG |
|---|---|---|---|
| CISS | CE | 94.37 | 94.48 |
| CISS | DR | 95.15 | 94.86 |
| UniT | DR | **97.21** | **98.71** |

Table 5: Comparison of certification ratio (%) in the Type II scenario.

the DR loss, in Yelp and AG, respectively. This is mainly because the calculation of $\hat{R}$ in UniT does not need IBP and is able to avoid the loose bound problem caused by IBP. Consequently, directly certifying in the BERT embedding space leads to more correctly classified samples that satisfy the certification condition. Note that the certified robust accuracy equals to the natural accuracy times the certification ratio. Thus, UniT can contribute to higher certified robust accuracy in the Type II setting, as it can have a higher ratio of test samples that satisfy the certification condition.

**Analysis on the Effect of the DR Loss.** Another aspect of the answer to Q2 is that the DR loss helps improve the robustness of the base model. To obtain a better understanding of the improvement achieved by the DR loss, w.l.o.g., we analyze how the DR loss improves the perturbed sample representation and classification in IMDB in the Type I scenario. We calculate the average $l_2$ norm between the representations of the perturbed and original samples, i.e., $||z' - z||$, learned from the CE loss and the DR loss in the test set, which are the inputs of the last FC layer. We also evaluate how the classifier (FC layer) is affected by calculating the margin change $\mathcal{M}(z') - \mathcal{M}(z)$ given perturbed samples.

| Measure | $||z' - z||$ | | $\mathcal{M}(z') - \mathcal{M}(z)$ | |
|---|---|---|---|---|
| Loss | CE | DR | CE | DR |
| Mean | 1.752 | **0.382** | -0.060 | **-0.032** |
| Variance | 1.374 | **0.494** | 0.877 | **0.282** |

Table 6: Comparison of representational $l_2$ norm and margin change between the CE and DR loss in IMDB.

For both of them, the closer to 0, the better. In Table 6, it is shown that the mean and variance of $||z' - z||$ obtained from the DR loss is significantly smaller than the ones obtained from the CE loss, which means that the DR loss effectively improves the representation compactness between perturbed samples and original ones. Besides, if we replace the CE loss with the DR loss, on average, the margin change will reduce from -0.060 to -0.032, given the perturbed samples

| Regularization | $\ell_{\hat{R}}$ | $\ell_{\hat{R}_{ub}}$ | $\ell_{\hat{R}_{lb}}$ |
|---|---|---|---|
| Natural Acc. (%) | 93.86 | 90.37 | **96.72** |
| Cert. Ratio (%) | 97.21 | **99.15** | 1.08 |
| CRA (%) | **91.24** | 89.60 | 1.05 |

Table 7: Trade-off between natural accuracy and certification ratio of UniT in Yelp.

in testing. Therefore, the DR loss can improve the quality of representation and certified results. To further demonstrate, we take one test sample as a visualization example for demonstration. Due to the space limit, we present the visualization result in Sec. A.9 in the Appendix.

**Analysis on Trade-Off Balance.** To answer Q3, we conduct the trade-off experiment in Yelp again but on UniT. Comparing Table 7 and Table 1, we can see that UniT does not suffer as much drop in certification ratio as CISS does when it adopts $\ell_{\hat{R}}$ compared to $\ell_{\hat{R}_{ub}}$, and $\ell_{\hat{R}}$ leads to the best performance. It is because UniT avoids the restriction caused by IBP. Therefore, the design of UniT for conducting certification in the embedding space can achieve a balance for the mentioned trade-off.

**Analysis on the Design of DR Loss.** For Q3, w.l.o.g., we further justify the design of the DR loss with ablation studies in IMDB in the Type I scenario. First, we illustrate the necessity of designing separate regularization terms for different modules. A straightforward choice that regularizes these two modules simultaneously is $||g(z') - g(z)||$, which is referred to as the adversarial logit pairing (ALP) [6], and $g(\cdot) \in \mathbf{R}^c$ is the corresponding logit vector. From Table 8, if we adopt ALP that penalizes two modules together, the CRA only improves from 85.36% to 86.64%. Thus, the decoupled regularization brings optimization flexibility.

| CE | ALP | $\ell_{fr}$ | $\ell_{cr}$ | CRA (%) |
|----|-----|-------------|-------------|---------|
| ✓ | | | | 85.36 (86.03) |
| ✓ | ✓ | | | 86.64 (86.36) |
| ✓ | | ✓ | | 88.08 (87.62) |
| ✓ | | | ✓ $(z+\epsilon)$ | 85.92 (87.42) |
| ✓ | | ✓ | ✓ $(z)$ | 87.68 (87.45) |
| ✓ | | ✓ | ✓ $(z')$ | 85.76 (85.71) |
| ✓ | | ✓ | ✓ $(z+\epsilon)$ | **89.04** (87.87) |

Table 8: Ablation study results on loss design in IMDB. $\ell_{fr}$ and $\ell_{fr}$ are indispensable. The natural accuracy is indicated by the parentheses.

Meanwhile, from Table 8, we can see that only using $\ell_{fr}$ or $\ell_{cr}$ causes a drop of 0.96% and 3.12%, respectively, compared to the DR loss. Thus, these two terms are indispensable for modular regularization. Additionally, we also validate our design of modeling perturbation for classifier by Gaussian noise. We incorporate the negative margin calculated from $z'$ with $\ell_{fr}$, but this regularization does not collaborate with the representational $l_2$ norm and worsens the regularization effectiveness. Thus, we should use $z+\epsilon$ to model the perturbation for the classifier module to enhance its robustness.

We further justify the design of each term in the MR term. For clarity, we here denote $t = \mathcal{M}(z+\epsilon)$. For the choice of the classifier regularization, it could be the ALP term, the margin loss $\max(-t, 0)$, or the function without $\max$ operation $-t$. Table 9 indicates that these choices all undermine the improvement brought by the representational $l_2$ norm. This is because the calculation of $||g(z') - g(z)||$ depends on $||z' - z||$ too, and it attaches too much

| Feature Extraction Reg. | Classifier Reg. | CRA (%) |
|-------------------------|-----------------|---------|
| $||z' - z||$ | $||g(z') - g(z)||$ | 85.44 |
| $||z' - z||$ | $\max(-t, 0)$ | 86.24 |
| $||z' - z||$ | $-t$ | 86.64 |
| $||z' - z||^2$ | $-\max(t, 0)$ | 87.36 |
| $||z' - z||$ | $-\max(t, 0)$ | **89.04** |

Table 9: Ablation study results on the regularization (Reg.) design in IMDB. We denote $t = \mathcal{M}(z+\epsilon)$.

weight to $||z' - z||$. Additionally, compared to $\max(-t, 0)$ or $-t$, the used $-\max(t, 0)$ has a selected gradient activation area for $t > 0$, so it adjusts the optimization direction when $z$ has not been trained well yet and sends a clearer optimization guidance than the others. We also note that the improvement decreases if we change the $l_2$ norm to the squared $l_2$ norm. Thus, the design in Figure 3 is reasonable.

## 5 Conclusion

The development of certified robust training in text is restricted by the lack of a unified framework and modular regularization. To alleviate these problems, we propose a unified framework named UniT to strengthen the certification of prediction results for text data. Under this framework, we introduce the DR loss combining the CE loss with the modular regularization term for different modules specifically to improve the base model robustness. Experiments show that UniT allows more correctly classified samples to satisfy the certification condition by avoiding IBP. Additionally, the DR loss can generate more similar representations for perturbed samples to the original ones and reduce the drop of prediction margin. Higher certified robust accuracy is achieved consequently.

## Acknowledgments and Disclosure of Funding

This work is partially supported by the National Science Foundation under Grant No. 1951729, 1953813, 2119331, 2212323, and 2238275.

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

# A  Appendix

## A.1  Limitations

The datasets used in experiments are mainly the ones used in the text classification task. We notice that there are other types of text datasets, such as question answering and text generation ones. In our experiments, we follow the existing works and test our methods in text classification datasets only. We expect future works to include a discussion on defending adversarial synonym substitution on text datasets of other types.

## A.2  Broader Impacts

The proposed UniT has a positive societal impact because it is able to generate not only accurate but also certified robust predictions on text-related tasks. Nowadays, large language models (LLMs) such as ChatGPT have attracted great attention due to their powerful performance and intriguing interface. However, they are mostly deployed as a black-box service for users to use and lack reliability for their predictions. For UniT, it can provide a robustness guarantee for predictions for tasks such as text classification, and it can be scalable for LLMs. As a result, UniT can positively impact the employment of LLMs in real-world applications and improve the trust between users and LLM service providers.

## A.3  Dataset Details

The used datasets are all in English. Each text sample is tokenized by the "BertTokenizer" provided by the Transformers [24] library. The detailed descriptions of the datasets are as follows:

1. IMDB [12] is a sentiment analysis dataset for movie reviews with either positive or negative sentiment.[†] Its text samples have comparatively longer lengths than the ones of the rest datasets. It has 25,000 train and test samples, respectively.

2. SST2 [20] is another binary text classification dataset for movie reviews. It has 67,349 train samples and 1,821 test samples.[‡] Its license is CC0.

3. Yelp [19] is a large-scale sentiment analysis dataset collected from restaurant reviews written by Yelp users with two classes, i.e., positive and negative ones.[§] It has 444,101 samples for training and 126,670 samples for testing. Its license is the Apache-2.0 license.

4. AG [29] is a comparably large-scale news classification dataset with 4 classes, including world, sports, science/technology and business.[¶] The number of train and test samples are 120,000 and 7,600, respectively.

## A.4  Calculation of Certified Robust Accuracy

Because the existing approaches use different ways to calculate the certified robust accuracy, to make a **fair comparison**, we follow the methods used in different scenarios and compare baselines separately.

- When comparing with **SAFER**, we follow the same setting that conducts sampling for choosing test samples and construct 5,000 perturbed samples for each test sample through random synonym replacement. The output from the smoothed model is derived by averaging the prediction of the 5,000 perturbed samples for each original text sample. We calculate the certified robust accuracy in this setting for both SAFER and UniT based on the certification condition proposed in Proposition 1 of SAFER [25]. The confidence level is set to 99.0%. We take a text sample in IMDB as an example and show one of the perturbed samples constructed by adversarial synonym substitution in Table 10.

---

[†]https://ai.stanford.edu/~amaas/data/sentiment/
[‡]https://www.kaggle.com/datasets/atulanandjha/stanford-sentiment-treebank-v2-sst2
[§]https://github.com/shentianxiao/language-style-transfer/tree/master/data/yelp
[¶]https://www.kaggle.com/datasets/amananandrai/ag-news-classification-dataset

| Sample | Text |
|---|---|
| Original Text (Label: Negative) | I went and saw this movie last night after being coaxed to by a few friends of mine. I'll admit that I was reluctant to see it because from what I knew of Ashton Kutcher he was only able to do comedy. I was wrong. Kutcher played the character of Jake Fischer very well, and Kevin Costner played Ben Randall with such professionalism. The sign of a good movie is that it can toy with our emotions. This one did exactly that. The entire theater (which was sold out) was overcome by laughter during the first half of the movie, and were moved to tears during the second half. While exiting the theater I not only saw many women in tears, but many full grown men as well, trying desperately not to let anyone see them crying. This movie was great, and I suggest that you go see it before you judge. |
| Perturbed Sample | I went and saw this cinematographic last nuit after being coaxed to by a few friends of mined. I'll admit that I was loath to see it because from what I knew of Ashton Kutcher he was only able to do comedy. I was awry. Kutcher played the character of Jake Fischer very well, and Kevin Costner played Ben Randall with such professionalism. The sign of a good cinematic is that it can plaything with unser emotions. This one did exactly that. The totaled theatres (which was sale out) was overcoming by laughter during the firstly half of the movie, and were moved to tears during the second half. While exit the theatres I not exclusively saw many daughter in tears, but varying full grown males as well, trying desperately not to letting anyone see them crying. This films was great, and I suggest that you go see it avant you judges. |

Table 10: Example of a perturbation sample. We color and underline the perturbed words (corresponding synonyms of original words) in the perturbed sample to demonstrate the difference between these two text samples.

- When comparing with **CISS**, we follow its setting of using the whole test set for calculating certified robust accuracy. The certified robust accuracy of CISS is calculated based on Algorithm 2 proposed in CISS [31], and we adopt the same certification process except that we adopt Theorem 1 as the certification condition for our unified framework in this setting. We show the pseudocode for prediction and certification under UniT in the Type II setting in Algorithm 1, and its details are discussed in Sec. A.7. Due to computing restrictions, the certification results are calculated from 9,000 perturbed samples constructed in the hidden space by adding Gaussian noise to the original sample embedding. The confidence level is set to 99.9%. During certification, the required inputs are the original text sample and the corresponding synonym set of each original word.

## A.5 Implementation

Since UniT is based on BERT, it has a similar parameter number to the one of BERT, which is 110M. We use the pretrained BERT model "bert-base-uncased" provided by the Transformers [24] library. When we conduct Type I training with UniT, for every dataset, we fine-tune the pretrained model with 3 epochs, which usually takes 10 minutes on an Nvidia A6000 GPU. When we conduct Type II training with UniT, the training takes about 48 hours for both datasets on an Nvidia A100 GPU. For Yelp and AG, we fine-tune the pretrained model with 110 and 200 epochs, respectively.

## A.6 Hyperparameters

The tuning of hyperparameters is not tricky for the DR loss due to their clear interpretation. During training with the DR loss, we set the hyperparameters $\nu = 0.1$ to keep the Gaussian noise relatively small, $\alpha = 0.7$ to allow the margin to increase while penalizing $l_2$ norm, and $\xi = 0.6$ to allow appropriate relaxation. In addition, while calculating the final loss, we set $\beta = 1$ to make the MR term and the CE loss have equal weight. In the Type II setting, the extra hyperparameters $\mu$ and $\gamma$ have been studied by [31], so we follow them to set $\mu = 1$ and incrementally increase $\gamma$ to 4 as the training epoch increases.

---

**Algorithm 1:** Prediction and certification by UniT in Type II Setting

---

**function** PREDICT

**Input**: Hard-label prediction function based on UniT $h$, Standard deviation of Gaussian noise $\sigma$, Embedding of original text $x$, Number of Gaussian noise $\eta$, Confidence level $\omega$

**Process**:

Draw $\eta$ samples of Gaussian noise, add them to $x$ repeatedly, and obtain a vector of class counts CNT for all perturbed inputs;

$\hat{y}_A, \hat{y}_B \leftarrow$ top two indices in CNT;

$\eta_A, \eta_B \leftarrow \text{CNT}(\hat{y}_A), \text{CNT}(\hat{y}_B)$;

**if** BINOMPVALUE$(\eta_A, \eta_A + \eta_B, 0.5) \leq \omega$ **then**

  Return $\hat{y}_A$; **else**

  | **Return** ABSTAIN

  **end**

**end**

**function** CERTIFY

**Input**: Hard-label prediction function based on UniT $h$, Standard deviation of Gaussian noise $\sigma$, Embedding of original text $x$, 1st number of Gaussian noise $\eta_1$, 2nd number of Gaussian noise $\eta_2$, Lower confidence $1 - \omega$

**Process**:

Draw $\eta_1$ samples of Gaussian noise, add them to $x$ repeatedly, and obtain a vector of class counts $\text{CNT}_1$ for all perturbed inputs;

$\hat{y}_A \leftarrow$ top index in $\text{CNT}_1$;

Obtain a vector of class counts $\text{CNT}_2$ similarly with $\eta_2$ samples of Gaussian noise;

$\underline{p_A} \leftarrow$ LOWERCONFBOUND$(\text{CNT}_2[\hat{y}_A], \eta_2, 1 - \omega)$;

**if** $\underline{p_A} > 0.5$ *and* $\hat{R} \leq \sigma \Phi^{-1}(\underline{p_A})$ **then**

| **Return** $\hat{y}_A$;

**end**

---

We also include the results of the influence of hyperparameters on the DR loss. Without the loss of generality, we test on the IMDB dataset in the Type I scenario. The results are obtained when we keep the values of other hyperparameters fixed as the ones we use.

**Influence of $\nu$.** From Table 11, a comparably small $\nu$ is beneficial to modeling the perturbation and loss optimization. Selecting a small $\nu$ as 0.05 has already increased the performance compared to that of using the CE loss. As $\nu$ grows greater than 0.1, the positive impact of using Gaussian noise to improve the robustness of the classifier module will gradually downgrade. Thus, a comparably small $\nu$ as 0.1 is most beneficial.

| $\nu$ | 0.05 | 0.1 | 0.15 | 0.2 |
|---|---|---|---|---|
| Result | 87.44 | 89.04 | 88.08 | 86.96 |

Table 11: Influence of $\nu$ on certified robust accuracy (%).

**Influence of $\alpha$.** $\alpha$ is the weight of the negative margin in the DR loss. As we have seen in the ablation study, the introduction of the negative margin contributes to enhancing the base model robustness by regularizing the robustness of the feature extraction and the classifier module. Thus, from Table 12, setting $\alpha$ comparatively high will be helpful for improving the certified robust accuracy. Thus, we can set $\alpha = 0.7$ to help improve the $l_2$ norm.

| $\alpha$ | 0.1 | 0.3 | 0.5 | 0.7 | 0.9 |
|---|---|---|---|---|---|
| Result | 88.08 | 88.72 | 88.24 | 89.04 | 89.04 |

Table 12: Influence of $\alpha$ on certified robust accuracy (%).

**Influence of $\beta$.** As the weight of the MR term, $\beta$ shall be set approximately equal to 1. From Table 13, we observe that only when $\beta$ is too large, e.g., $\beta = 2$, can the MR term damage the training of the base model. Therefore, we can just set $\beta$ to have equal weights with the CE loss to improve the robustness of the base model.

| $\beta$ | 0.5 | 1 | 1.5 | 2.0 |
|---|---|---|---|---|
| Result | 88.40 | 89.04 | 88.24 | 83.60 |

Table 13: Influence of $\beta$ on certified robust accuracy (%).

**Influence of $\xi$.** From Table 14, the introduction of relaxation will be helpful for getting high certified robust accuracy. As shown in Table 14, the certified robust accuracy grows as $\xi$ increases from 0 to 0.6 and gradually decreases when $\xi$ gets higher. Therefore, 0.6 is the relaxation hyperparameter we use in our experiment.

| $\xi$ | 0 | 0.2 | 0.4 | 0.6 | 0.8 | 1.0 |
|---|---|---|---|---|---|---|
| Result | 86.40 | 86.96 | 87.92 | 89.04 | 88.34 | 87.44 |

Table 14: Influence of $\xi$ on certified robust accuracy (%).

### A.7 Proof for Theorem 1

**Proof.** Given the results of Theorem 1 in [1], for all $||\Delta|| < R$, we have $h(x + \Delta) = y$. Denote $u$ the word embedding function. Recall that $x = [u(w_1), \cdots, u(w_n)]$ for the text sample $X = [w_1, \cdots, w_n]$, and $x' = [u(w'_1), \cdots, u(w'_n)]$ for any allowed perturbed sample $X' = [w'_1, \cdots, w'_n] \in A(X)$. Because each word is independent of the others in the embedding space, the $l_2$ norm between $x$ and any $x'$ is

$$||x' - x|| = \sqrt{\sum_{i=1}^{n} ||u(w_i) - u(w'_i)||^2}. \tag{8}$$

Note that for each word $w_i \in X$, the embedding set of its synonyms is $U_i = \{u(w_i^{(1)}), \cdots, u(w_i^{(m_i)})\}$, and the maximum deviation between $w_i$ and any word $w'_i \in S(w_i)$ is $||v_i|| = \max_{e \in U_i} ||u(w_i) - e||$. Since each word is independent of the others, the maximum deviation of $||x' - x||$ caused by all possible combinations of the synonyms of different words is

$$\hat{R} = \max ||x' - x|| = \sqrt{||v_1||^2 + \cdots + ||v_n||^2}. \tag{9}$$

Now since we have $\hat{R} \leq R$, we correspondingly have $||x' - x|| \leq R$ for all $X' \in A(X)$, thus

$$h(x') = h(x + (x' - x)) = h(x) = y, \tag{10}$$

for any $X' \in A(X)$ whose corresponding text embedding is $x'$. Q.E.D.

**Remark.** In certification, we also follow the calculation used by [1] and [31] that $R$ is lower bounded by $\sigma\Phi^{-1}(\underline{p_y})$, where $\underline{p_y}$ is the lower bound of $\mathbb{E}_\delta[h(x + \delta) = y]$ estimated from the Binomial proportion confidence interval. Thus, the certification condition becomes $\underline{p_y} > 0.5$ and $\hat{R} \leq \sigma\Phi^{-1}(\underline{p_y})$, which is harnessed in [1] and [31] as well.

We also show the prediction and certification process in the Type II setting in Algorithm 1. This process mainly follows the same idea as those of [1] and [31]. In Algorithm 1, BINOMPVALUE($\eta_A, \eta_A + \eta_B, p$) returns the p-value of the two-sided hypothesis test that $\eta_A \sim$ Binomial($\eta_A + \eta_B, p$). And LOWERCONFBOUND($\kappa, \eta, 1-\omega$) returns a one-sided $(1-\omega)$ lower confidence interval for the Binomial parameter $p$ given a sample $\kappa \sim$ Binomial($\eta, p$).

### A.8 Geometric Interpretation of MR Term

Eq. (6) has an explicit geometric interpretation. As shown in Figure 4, after projecting the original sample and perturbed sample in the high-dimensional representation space $\mathbf{R}^d$ (the input space of the last FC layer $g$), Eq. (6) requires the perturbed representation $z'$ to locate around $z$ within the radius $r = \xi + \alpha \cdot \mathcal{M}(z + \epsilon)$. That is, it tries to guide the original sample and perturbed sample representations to be close to each other and improve the inter-sample compactness of the high-dimensional space with the margin information.

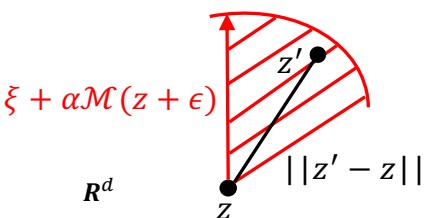

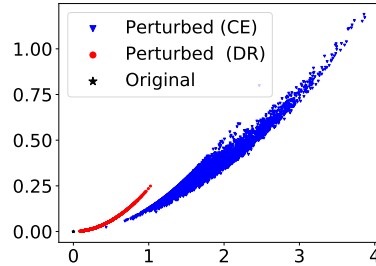

Figure 4: Geometric interpretation of the MR term.

Figure 5: Comparison of perturbed text sample representation distribution. The representations of perturbed samples obtained from DR loss are closer to the original ones than the ones obtained from CE loss.

## A.9 Visualization of Representation Distribution

As mentioned in the Experiment section, we take one test sample as a visualization example to demonstrate that DR loss is able to improve the representation compactness between perturbed samples and the original ones. The used text sample is randomly chosen from the IMDB dataset and the visualization experiment is conducted in the Type I scenario. We now show the visualization result as follows.

Given this randomly chosen text sample, in the certification process, we will need 5,000 perturbed samples to certify the prediction result. As Figure 5 shows, for all the used perturbed samples for certification, we first obtain two groups of representations with the CE and DR loss from the feature extraction module, respectively. We denote the $l_2$ distance between the original sample representation $z$ and the representation of any perturbed sample $z'$ as $L$. We then project those high-dimensional representations $z'$ into a two-dimensional space based on $L$: if the $l_2$ distance and angle of $z'$ with $z$ is $L$ and $\omega$ respectively, the coordinate of $z'$ in Figure 5 is $(L\cos\omega, L\sin\omega)$, and the origin $(0,0)$ represents $z$. In this example, the average $||z' - z||$ for representations obtained from the CE and DR loss are 1.768 and 0.353, respectively. From the visualization in Figure 5, the perturbed sample representation obtained from the DR loss distributes much closer to $z$ with a smaller divergence, which demonstrates that the representation learned from the DR loss is of higher quality.

