# OpenReview forum: "UniT: A Unified Look at Certified Robust Training against Text Adversarial Perturbation"
_NeurIPS.cc/2023/Conference — NeurIPS 2023 poster_

### Official Review · Reviewer_yTLz · 2023-06-27

**Soundness:** 4 excellent
**Presentation:** 3 good
**Contribution:** 2 fair
**Rating:** 6
**Confidence:** 4

**Summary:**

The paper presents UniT, a unified certified robust training against text adversarial perturbation. The paper identifies two frameworks of robust training: type (I) does random data augmentation over the text inputs, and type (II) adds Gaussian noise to the latent feature space. The paper unifies two types of training by doing data augmentation in the inputs and adding Gaussian noise to the latent feature space. The paper also introduces a novel loss function, which modularizes the feature extraction regularization and robust regularization as a loss term to fuse these two types of training besides the original cross-entropy loss.

The paper evaluates UniT and demonstrates that it consistently outperforms SAFER and CISS. The paper also does ablation studies of different loss function designs and different hyper-parameter settings (in the appendix).

**Strengths:**

1. The modular loss function design is interesting and novel. The ablation studies of the loss function are very strong.
2. The paper outperforms existing certified approaches SAFER and CISS.

**Weaknesses:**

1. The paper only handles the perturbation space of synonym substitutions but not other text adversarial perturbations such as insertion, deletion, and their combinations. This weakness makes the title "text adversarial perturbation" overclaimed.
2. The comparison between UniT and CISS needs to be presented more clearly. See questions 2-4.

**Questions:**

Comment:
1. It would be better to also present natural Acc in Table 7 and 8. And in all the hyper-parameter tables.
2. The motivation part in lines 202-208 is not well-justified. I think the IBP is necessary if one wants to get a certification. The reason why UniT can get rid of IBP but still gets a certification is because UniT uses BERT embedding and add noise before feature extraction. Then IBP in CISS only needs to analyze the embedding part, which is easy to analyze and in fact, do not need IBP at all.
3. In lines 283-285, the paper states "which will not regularize g if z is well learned". However, it is not the case, the only thing the -max(., 0) does is to not penalize the wrongly classified cases. In other words, "which will not regularize g if z is NOT well learned".

Questions:
1. Is there any intuition why the paper chooses the l2 norm over cosine similarity?
2. The original $l_{\hat{R}}$ loss used in type II training also tries to minimize the distance of $\| z-z'\|$ in the latent space of IBP, no?
3. What's the performance of CISS using their original loss term in Table 3? Table 3 only shows the results of CE and DR loss.
4. Why can CISS not do IBP on BERT? UniT and CISS both add Gaussian noise before feature extraction. Then CISS only needs to analyze the BERT embedding part, which is easy because of the independence of embedding of each word, as mentioned in line 231. I agree that IBP might be looser, consider x=[0,0], x_1=[0,1], x_2=[1,0], then $\hat{R} = 1$ but the over-approximated $\hat{R}$ via IBP should be $\sqrt{1+1}=\sqrt{2}$.

**Limitations:**

The authors do not address the limitation. One limitation of the paper is that the paper only handles word substitutions but not general perturbation spaces such as insertion, deletion, and their combinations.

---

> ### Author Rebuttal · Authors · 2023-08-10
>
> We would like to sincerely express our thankfulness to the reviewer for the positive and insightful feedback on our paper. We are deeply appreciative of the acknowledgment of our paper on the novelty of loss design and the practical effectiveness of the proposed UniT framework. We take every comment or question seriously and hope that our response can clarify the concerns. Meanwhile, we will be more than happy to address any additional concerns.
>
>
> **Q1: Is there any intuition why the paper chooses the l2 norm over cosine similarity?**
>
> **A**: UniT chooses l2 norm over cosine similarity in Eq. (4) because l2 norm can limit the influence of perturbation on downgrading the prediction margin (defined in Sec. 2.2) and is good for certification. For the ease of illustration, we confine our discussion in a binary classification case and the (unbiased) weight of the last layer is for classifying the original input vector $z$ into class $a$ and class $b$ are $w_a$ and $w_b$, respectively. Suppose the ground truth is class $a$. Thus, the prediction margin will be $M(z)=w_a^T z-w_b^T z=(w_a-w_b)^T z$. Similarly, the prediction margin given the perturbed representation in $z^\prime$ will be $M(z^\prime)=(w_a-w_b)^T z^\prime$. Thus, the influence on the prediction margin after the attack will be $M(z^\prime)-M(z)= (w_a-w_b)^T (z^\prime-z)$. Since we want to reduce this influence for better certification results, we shall use the l2 norm of $z^\prime-z$ as an optimization target to limit the change $ M(z^\prime)-M(z)$. We will include this discussion in the final version.
>
> **Q2: Does the original $l_{\hat{R}}$ loss used in type II training also try to minimize the distance of $||z−z^\prime||$ in the latent space of IBP?**
>
> **A**: No, it doesn’t. $z$ and $z^\prime$ are the output from the penultimate layer (the last but one layer), while $l_{\hat{R}}$ in the Type II training is calculated from the mid-level layer. To specify, $l_{\hat{R}}$ uses the feature output from the feature encoder that is before the base model, and the base model is the module that outputs $z$ and $z^\prime$. Therefore, the original $l_{\hat{R}}$ cannot minimize $||z−z^\prime||$.
>
> **Q3: What's the performance of CISS using their original loss term in Table 3?**
>
> **A**: To clarify, the performance of CISS using their original loss terms is the first row in Table 3. We use CE to denote the loss of CISS because the main difference between our loss and the loss of CISS lies in the CE term, and both losses include $l_{\hat{R}}$. We have discussed this difference in L343-L345 and illustrated our loss terms for Type II in L294-L297.
>
> **C2 and Q4: On the necessity of IBP in CISS.**
>
> **A**: We believe C2 and Q4 is correlated and would like to answer them together. Recall that in the context of certified robustness there are two important things for obtaining high certified robust accuracy: training and certification. Thus, for this research task, the design of the training mechanism is related to or depends on the design of the certification condition. CISS designs a certification condition in the mid-level latent space, so it has to introduce IBP for the purpose of certification. Since the parameters in IBP has to be learned, it needs to be trained with the base model together to make the extracted features satisfy the certification condition (shown in L148). Thus, IBP cannot be removed from CISS unless CISS has a certification condition not related to the latent space. And in fact, one of our main contributions in this paper is to propose a stronger certification condition by taking into consideration the independence of words in the embedding space, and this certification condition helps us design our Type II training without IBP.
>
> **C1: It would be better to also present natural Acc in Table 7 and 8. And in all the hyper-parameter tables.**
>
> **A**: We thank the reviewer for the suggestion. We also have natural accuracy in the record and will present them in the final version. For example, Table 7 added with natural accuracy will be as follows. The natural accuracy is calculated on all test samples.
>
> Table R1: Ablation study results on loss design in IMDB.
> CE | ALP | $l_{fr}$| $l_{cr}$ | Natural Acc (%) | CRA (%)
> ---|---|---|---|---|---
> $\checkmark$  |  |  |   |   86.03 |  85.36
> $\checkmark$ | $\checkmark$ | |   |  86.36 |  86.64
> $\checkmark$ |   |  $\checkmark$ |   | 87.62  |  88.08
> $\checkmark$ |  |    |  $\checkmark$  (Input: $z+\epsilon$) | 87.42 | 85.92
> $\checkmark$ |   | $\checkmark$ |  $\checkmark$ (Input: $z$)  | 87.45  |    87.68
> $\checkmark$ |   |$\checkmark$ | $\checkmark$ (Input: $z^\prime$) |  85.71 |   85.76
> $\checkmark$ |   | $\checkmark$ | $\checkmark$  (Input: $z+\epsilon$)  |  87.87 | 89.04
>
> **C3: A typo in L285.**
>
> **A**: Thank you for your carefulness when reading our paper. You have understood our loss design correctly and it should be "which will not regularize g if z is NOT well learned". We will correct this typo in the final version.
>
> **C4: The paper only handles the perturbation space of synonym substitutions but not other text adversarial perturbations such as insertion, deletion, and their combinations.**
>
> **A**: Throughout our literature review on text adversarial perturbations, synonym substitution has been considered as the standard way to construct text adversarial perturbation in adversarial attack and defense. This is because insertion and deletion have been regarded as two special forms of word substitution: insertion can be considered as replacing an ‘[EOS]’ token with a new word and putting the replaced word in the inserted position, and deletion can be considered as replacing a word by the '[UNK]' token. Thus, previous literature holds that the ability to defend against synonym substitution generally reflects the ability to defend against other two forms of adversarial perturbations. Thus, we follow the existing setup to conduct adversarial perturbations in the context of synonym substitutions.

---

> > ### Comment · Reviewer_yTLz · 2023-08-12
> > **Response to authors**
> >
> > Thanks for the efforts and detailed response. The response answers all my questions. Currently, I think that this paper should be accepted. I will read other reviews and corresponding response in the next few days.

---

> > > ### Author Response · Authors · 2023-08-13
> > >
> > > Dear Reviewer yTLz,
> > >
> > > Thank you for the encouraging words! We deeply appreciate your acknowledgment of the high quality of our paper!
> > >
> > > Sincerely,
> > >
> > > Authors

---

### Official Review · Reviewer_3SMy · 2023-07-04

**Soundness:** 3 good
**Presentation:** 3 good
**Contribution:** 3 good
**Rating:** 5
**Confidence:** 2

**Summary:**

This paper proposes a unified framework called UniT for certified robust training against text adversarial perturbation. UniT can train and certify in both discrete word space and continuous latent space by working in the word embedding space, without extra modules or loose bounds. It also introduces a decoupled regularization loss called DR loss to improve the base model robustness by regularizing the feature extraction and classifier modules separately. The paper shows that UniT outperforms existing state-of-the-art methods in both scenarios, and DR loss enhances the certified robust accuracy by providing modular regularization.

**Strengths:**

The paper clearly identifies the limitations of existing approaches and highlights the advantages of combining them with a novel DR loss into a unified framework. The authors have made an effort to address these limitations and have proposed a novel solution that capitalizes on the strengths of both discrete word space and continuous latent space approaches. Its evaluation demonstrates the effectiveness of their proposed approach.

**Weaknesses:**

1. The selection of synonyms (Type I) is indeed an important aspect of the paper, and it would be helpful if the authors could provide more clarity on how synonyms are chosen based on embeddings. I'm not sure about the probability of changing labels due to synonym substitution for the task of sentiment analysis. Could the authors provide some concrete examples? Additionally, a discussion on how the base model is obtained (whether it's fine-tuned BERT or not) and any improvements in generalization performance could shed more light on the robustness of the proposed approach.

2. The paper's focus on BERT and sentiment classification may indeed limit its applicability to other tasks or models, especially in the era of large-scale pre-trained models with improved generalization capabilities. The authors could address this concern by discussing the potential of their framework to be extended to other tasks and models, and whether the problem they are investigating remains relevant in the current research context.

3. The paper mainly discusses synonym substitution and noise addition, but there are now more advanced perturbation methods based on large language models (LLMs) that could potentially generate more realistic adversarial examples (e.g., through rephrasing or prompting). A comparison or discussion of these alternative methods and their implications for the proposed approach could provide a more comprehensive understanding this paper's contribution.

**Questions:**

The introduction of DR loss indeed results in a more unified model, but it also brings additional hyperparameters. The paper could provide guidance on how to efficiently and accurately select the best hyperparameters for practical applications. This would make the proposed approach more accessible and easier to implement for researchers and practitioners alike.

**Limitations:**

See the weakness section

---

> ### Author Rebuttal · Authors · 2023-08-10
>
> We genuinely thank the reviewer for the positive and constructive feedback on our paper. We thank the reviewer for recognizing the novelty and effectiveness of our solution. Each comment and question is of great importance to us, and we hope our response can address the raised concerns. Moreover, we are eager to answer any additional questions.
>
> **Q1: On the guidance for selecting the best hyperparameters.**
>
> **A**: This is a good question, and in fact, we have provided guidance on selecting hyperparameters in Section A.6 in the submitted Appendix. Due to the word limit, we kindly ask the reviewer to view our experimental results on the influence of hyperparameters in the Appendix. From those results, we can find that the tuning of hyperparameters is not tricky for the DR loss due to their clear interpretation, and these settings can be used widely for different datasets. To specific, $\mu$ is kept as small as 0.1 to keep Gaussian noise relatively small. $\alpha$ is set to 0.7 to ask the margin to increase while penalizing $l_2$ norm of text representation. $\epsilon=0.6$ is fine for allowing some relaxation. And we just make CE loss and MR term have equal weights by setting $\beta=1$.
>
> **C1: On the selection of synonyms, their influence on prediction labels, and how the base model is obtained.**
>
> **A**: We would like to provide those implementation details as follows. With respect to synonym selection, in Type I scenario we follow the baseline SAFER and use the same set of synonyms and the selection of synonyms is based on the similarity between two words in a pretrained embedding space. To illustrate, by using the embedding space of GLOVE embeddings (Pennington et al., 2014), for a specific word, any other word that has a cosine similarity that is greater than 0.8 are considered a synonym of that word.
>
> As for how synonym substitution can change the label, we would like to show one example in the Yelp dataset. Given a text sample “Great buffet. Lots of selections. The prime rib was delicious. It was worth the 30 dollars” that is correctly predicted as a sample with positive sentiment by a text classifier, the prediction can be changed to negative sentiment when only the word “delicious” is changed by “loverly”. That is, “Great buffet. Lots of selections. The prime rib was loverly. It was worth the 30 dollars” will be classified incorrectly by the text classifier when only one word is changed. We also include a long text sample in Table 9 in the Appendix for illustrating how synonym substitution changes the prediction results. Since our work mainly focuses on certified robustness and not on text adversarial attacks, and due to the word limit, we kindly refer the reviewer to representative works on text adversarial attacks such as TextBugger (Li et al., 2019), and TextFooler (Jin et al., 2019) if needed to get a detailed look on how synonym substitution changes prediction labels and fool text classifiers.
>
> Finally, the base model is obtained from pre-trained “bert-base-uncased” BERT provided by the transformers package by huggingface. We have included this discussion in Section A.5 in the Appendix and kindly ask the reviewer to review that section if further information is needed due to the word limit.
>
> **C2: On the generalization capabilities of the proposed method.**
>
> **A**: We thank the reviewer for this question. As mentioned in L160-L163, our focus on BERT and sentiment classification is mainly because of the existing literature related to the task of prediction certification for text data. Our structural decomposition analysis of the base model in Figure 2 is general among different models, such as RoBERTa, ALBERT and even more recently proposed foundation models such as ViLT and UNITER. Since the proposed DR loss is designed from this general point of view, DR loss can still be used for those latest structures, and it can help improve the robustness of those models in other classification tasks such as visual-question answering. We will include this discussion in the final version.
>
> **C3: On the other ways of generating adversarial examples such as through rephrasing or prompting.**
>
> **A**: We have also noticed the methods of generating adversarial examples by rephrasing and prompting, but they have the weakness of changing a lot of words and increasing the perturbation rate, which disobeys the definition of good adversarial examples that the difference between adversarial examples and the original samples should be imperceptible. For example, the method proposed by (Qi et al., 2021) rewrites the text with a different style by rephrasing and the changes are easily noticeable. In this sense, the adversarial examples generated in those ways are not of high quality because of their high perturbation rates. We thank the reviewer for the suggestion and will include those works in the discussion to distinguish why we shall focus on the adversarial examples conducted by synonym substitution.
>
> **References**
>
> Pennington, J., Socher, R., & Manning, C. D. (2014, October). Glove: Global vectors for word representation. In Proceedings of the 2014 conference on empirical methods in natural language processing (EMNLP) (pp. 1532-1543).
>
> Li, J., Ji, S., Du, T., Li, B., & Wang, T. (2019, January). TextBugger: Generating Adversarial Text Against Real-world Applications. In 26th Annual Network and Distributed System Security Symposium.
>
> Jin, D., Jin, Z., Zhou, J. T., & Szolovits, P. (2019). Is bert really robust? natural language attack on text classification and entailment. arXiv preprint arXiv:1907.11932, 2, 10.
>
> Qi, F., Chen, Y., Zhang, X., Li, M., Liu, Z., & Sun, M. (2021). Mind the style of text! adversarial and backdoor attacks based on text style transfer. arXiv preprint arXiv:2110.07139.

---

> > ### Comment · Reviewer_3SMy · 2023-08-19
> >
> > Thank you for your response, which has partially allayed my concerns. While I remain unconvinced that the proposed training method will become a predominant solution in the near future, I acknowledge the paper's merits in terms of concept and overall quality. Consequently, I maintain my initial score and lean towards accepting this paper.

---

> > > ### Author Response · Authors · 2023-08-19
> > >
> > > Dear Reviewer 3SMy,
> > >
> > > We thank you for your positive feedback on our paper and the acknowledgment of the strength of our paper in terms of concept and overall quality!
> > >
> > > Sincerely,
> > >
> > > Authors

---

### Official Review · Reviewer_Wx5f · 2023-07-06

**Soundness:** 3 good
**Presentation:** 3 good
**Contribution:** 3 good
**Rating:** 6
**Confidence:** 4

**Summary:**

This paper focuses on the certified robustness of language models. To improve the certified robustness, the authors propose a better robust training method that enables robust feature extraction and a larger prediction margin. Experiment results show the effectiveness of the proposed DR loss, leading to a better certified accuracy compared to traditionally used CE loss.

**Strengths:**

This paper is well-motivated. The authors first identify the existing challenge of certification when using CE loss. Based on that, the DR loss is proposed to alleviate the non-robust feature extraction and improve the prediction margin. Empirical evaluations also demonstrate the effectiveness of the proposed DR loss.

**Weaknesses:**

- An important baseline is missing: robust training. The overall objective of the proposed DR loss is to improve the robustness of the base classifier so that it is easier to be certified. Robust training such as min-max adversarial training [1], and TRADES [2] is also known to be beneficial for the certification. Since the proposed method manipulates the training objective, which is very similar to min-max adversarial training and TRADES, it would be great to compare the certification performance of the model after training with [1,2] and the proposed loss. For example, one may use greedy search-based attack methods from  [3] to find the perturbation for each batch during training. Then minimize the CE loss on the perturbed batch (when using TRADES it will be slightly different). Please note that the synonym set for each word should be consistent with SAFER's (if compare in the Type-I scenario) rather than using the original candidate set of the attack methods.
- An empirical robustness evaluation would make the evaluation more comprehensive. It would greatly demonstrate the effectiveness of the proposed method if improvements in the empirical robustness can be observed and outperform robust training methods.

Overall, this paper is well-motivated and clearly written. The proposed technique makes sense and is verified to be effective in some settings. If my concerns could be addressed, I would like to raise my rating.



[1] Madry, Aleksander, et al. "Towards deep learning models resistant to adversarial attacks." arXiv preprint arXiv:1706.06083 (2017).

[2] Zhang, Hongyang, et al. "Theoretically principled trade-off between robustness and accuracy." *International conference on machine learning*. PMLR, 2019.

[3] Morris, John X., et al. "Textattack: A framework for adversarial attacks, data augmentation, and adversarial training in NLP." *arXiv preprint arXiv:2005.05909* (2020).

**Questions:**

The maximum deviation caused by all synonyms computed in Equation (1) seems to make the $l_{2}$ the constraint for the perturbation too large. Assuming the average  $l_{2}$ radius for each token embedding is $r$, then the total $l_{2}$ radius would be enlarged by $\sqrt{n}$ times.  The input length of some datasets used in this paper could be rather large. For example, IMDb contains average 300-500 words. After tokenization, the number of tokens, $n$, could be more than 500. Then the radius of the perturbation is enlarged more than 10x. In that case, the certification process in Theorem 1 (Equation 3) would be very possible to lead to a 0 certified accuracy. I would ask for more details about the computation of $\hat{R}$ and additional certification operations to ensure the validity of the results.

**Limitations:**

The authors discussed the limitations and potential negative social impact of their work

---

> ### Author Rebuttal · Authors · 2023-08-10
>
> We sincerely thank the reviewer for the encouraging and constructive comments. We are deeply grateful to the reviewer for acknowledging the significance of our research and thinking highly of the novelty and effectiveness of the proposed DR loss. We take the posted comments and questions seriously and would like to address the concerns of the reviewer on the comparison with robust training loss and empirical robustness evaluation with the following response. We will be more than happy to address any further concerns.
>
> **C1: An important baseline is missing: robust training.**
>
> **A**: We thank the reviewer for the comment and would like to further demonstrate the effectiveness of the designed loss. Without loss of generality, we use Type I scenario for illustration and extend our experiments in Table 1 with the following comparison results. We use the representative PGD (Madry et al., 2017) loss and TRADES (Zhang et al., 2019)  loss to adversarially train the model with the perturbation added to the word embedding space, which are the robust training baselines.
>
> Table R1: Comparison of certified robust accuracy (%) with robust training loss in the Type I scenario.
> Base Model | Loss | IMDB| SST2 | Yelp | AG
> ---|---|---|---|---|---
>  BERT | CE Loss|85.36 |91.65| 97.19| 93.78
>  BERT | PGD Loss | 87.52 | 90.28| 97.86 | 93.98
>  BERT | TRADES Loss | 86.80 | 90.44 | 97.56 | 93.96
>  BERT | DR Loss | **89.04**  | **93.02** | **97.87**  |**94.31**
>
> From Table R1, we can find that DR loss consistently outperforms PGD loss and TRADES loss in four datasets. The increment of certified robust accuracy (CRA) is relatively considerable for the datasets with longer text such as IMDB and SST2. Specifically, the CRA of DR loss is 1.52% and 2.24% higher than those of PGD and TRADES in IMDB, respectively, and it is 2.74% and 2.58% higher than those of PGD and TRADES in SST2, respectively. Thus, DR loss is more suitable for the certified robust training task because (1) the perturbation for each word is randomly drawn from their corresponding synonyms, which covers a larger variety of perturbed samples, and (2) our designed modular regularization eyes on reducing the negative effect of the perturbation on prediction margin, which is the key to satisfying certification condition. We will include this discussion in the final version.
>
> **C2: An empirical robustness evaluation would make the evaluation more comprehensive.**
>
> **A**: We thank the reviewer for the comment and would like to address this concern with the following results on empirical robustness evaluation. Without loss of generality, we conduct a comparison of empirical robust accuracy with PGD loss and TRADES loss on the four datasets. We use BERT as the base model, and train BERT with PGD loss, TRADES loss, and the proposed DR loss, respectively. After that, we follow the existing adversarial attack setup and randomly draw 1,000 test samples for adversarial example generation. The adversarial examples are generated by TextBugger (Li et al., 2019), a representative text adversarial method. The results are as follows.
>
> Table R2: Comparison of empirical robust accuracy (%) with robust training loss. We also show the natural accuracy which is indicated by the parentheses.
>
> Base Model | Loss | IMDB| SST2 | Yelp | AG
> ---|---|---|---|---|---
>  BERT | PGD Loss | 62.0 (87.1) | 87.6 (91.5) | 92.4 (96.9)  | 85.5 (93.7)
>  BERT | TRADES Loss | 57.8 (84.8) | 86.3 (91.7) | 91.9 (97.0) | 84.4 (94.3)
> BERT | DR Loss | **72.7** (86.9)  | **89.8** (93.3) | **96.9** (98.4) | **87.6** (92.9)
>
> From the comparison results in Table R2, the proposed DR loss in UniT also outperforms PGD loss and TRADES loss with respect to empirical robust accuracy. This also attributes to the individual robustness supervision provided by DR loss for each module, which helps each module to work together more coherently to defend against adversarial attacks. We will include this discussion in the final version.
>
>
>
> **Q1: On the computation of $\hat{R}$ and additional certification operations to ensure the validity of the results.**
>
> **A**: This question is mainly about the certification process in Type II and especially the calculation of $\hat{R}$. It is a good question for $\hat{R}$ is the key to certification in Type II scenario. To answer this question, firstly, it is true that some datasets generally have long text inputs, and we follow previous works and truncate the text inputs for efficiency reasons. For example, the input text of IMDB is truncated to 128 tokens in training and certification. Secondly, the embedding of each word is not frozen during training, i.e., they are regarded as part of model parameters. Since in Type II scenario, the word embedding is supervised with a regularizer $\ell_{\hat{R}}$ (shown in L295), the embedding of each word tends to shrink its norm and becomes small. As a result, in certification, $\hat{R}$ will be relatively small after training, which makes the satisfaction of certification in Theorem 1 much easier.
>
> **References**
>
> Madry, A., Makelov, A., Schmidt, L., Tsipras, D., & Vladu, A. (2017). Towards deep learning models resistant to adversarial attacks. arXiv preprint arXiv:1706.06083.
>
> Zhang, H., Yu, Y., Jiao, J., Xing, E., El Ghaoui, L., & Jordan, M. (2019, May). Theoretically principled trade-off between robustness and accuracy. In International conference on machine learning (pp. 7472-7482). PMLR.
>
> Li, J., Ji, S., Du, T., Li, B., & Wang, T. (2019, January). TextBugger: Generating Adversarial Text Against Real-world Applications. In 26th Annual Network and Distributed System Security Symposium.

---

### Official Review · Reviewer_FiMX · 2023-07-09

**Soundness:** 3 good
**Presentation:** 3 good
**Contribution:** 3 good
**Rating:** 6
**Confidence:** 3

**Summary:**

This research paper discusses the vulnerability of Deep Neural Networks (DNNs) used in Natural Language Processing (NLP) tasks against adversarial attacks, specifically word-level adversarial perturbation (or synonym substitution). The authors delve into two existing training frameworks (Type I and Type II) for these NLP models, highlighting the shortcomings related to unified training frameworks and the robustness of the base model. To overcome these limitations, the authors suggest a novel framework, UniT, which merges the two types of models to provide stronger certified robustness. They also propose a Decoupled Regularization (DR) loss to optimize robustness regularization for individual modules. Experimental results provided deliver evidence that the UniT with DR loss improves the certified robust accuracy of both types of certification scenarios.

**Strengths:**

1. **Novelty**: The paper presents an original perspective on certified robust training for adversarial attacks in text data bringing unique insights and implementations, which addresses the identified gaps in the field.
2. **Improved Certification Accuracy**: The proposal of a unified framework (UniT) and a novel decoupled regularization (DR) loss show promising results, achieving higher certified robust accuracy. This moves us towards creating models with stronger robustness.
3. **Bypasses IBP Issues**: The UniT framework allows Type II methods to bypass using Interval Bound Propagation (IBP) during training, which has been shown to have problems in certification due to its loose bound problem. This successfully solves a major complexity in the training process.

**Weaknesses:**

1. **Increased Complexity**: While the UniT framework and DR loss may improve robustness, they potentially increase the complexity of model training because it requires handling the embedding space as an intermediate for unification and decoupling the CE loss, which could be time and resource consuming.
2. **Limited Validation**: Although the paper claims improved results, tests and validations seem limited. It would be beneficial to test the proposed methods on different tasks or datasets to better assess their efficacy and robustness.

**Questions:**

- No particular questions

**Limitations:**

yes

---

> ### Author Rebuttal · Authors · 2023-08-10
>
> We sincerely thank the reviewer for the positive feedback and comments, and we appreciate the reviewer very much for the recognition on the novelty, improved performance, and meaningfulness of our solution. Taking the raised questions seriously, we answer each of them with utmost sincerity as follows. We genuinely hope our response can clarify the concerns and will be more than happy to address any further concerns.
>
> **C1: Increased Complexity.**
>
> **A**: We admit that compared to the training using CE loss, calculating the proposed DR loss needs to increase the computation complexity due to the calculation of getting the features of perturbed input and original input during training. Such a tradeoff is natural: to obtain better model robustness we have to increase computation cost. In fact, the increase of computation cost also occurs for existing robust training techniques such as using PGD (Madry et al., 2017) loss and TRADES (Zhang et al., 2019) loss for improving empirical adversarial robustness. Additionally, we want to highlight the fact that compared to existing robust training techniques, the computation cost of calculating DR loss is relatively lower. Suppose the calculation cost of CE loss is $C$. The computation cost of DR loss would be approximately $2C$. However, for existing robust training techniques such as using PGD loss and TRADES loss, suppose the number of steps of generating adversarial examples is $k$, and the computation cost of PGD loss or TRADES loss will be $k\cdot C$. Since $k$ is generally larger than 2 in their implementation, the computation cost of DR loss is relatively smaller. We will include this discussion in the final version.
>
> **C2: Limited Validation.**
>
> **A**: We thank the reviewer for the suggestion and introduce the empirical robust training methods including PGD loss and TRADES loss as baselines to further validate the effectiveness of the proposed methods. Firstly, we want to compare DR loss with the added baselines on certified robust accuracy. Secondly, we conduct a comparison between ours and those baselines in terms of empirical robust accuracy. We will include those results in the final version, which further demonstrates the effectiveness of the proposed DR loss in improving model robustness. To detail:
>
> Firstly, without loss of generality, we extend our experiments in Table 1 in Type I scenario by adding PGD loss and TRADES loss as baselines. The results are shown in Table R1.
>
> Table R1: Comparison of certified robust accuracy (%) in the Type I scenario.
>  Base Model | Loss | IMDB| SST2 | Yelp | AG
> ----|--|---|---|---|---
>  BERT | CE Loss|85.36 |91.65| 97.19| 93.78
>  BERT | PGD Loss | 87.52 | 90.28| 97.86 | 93.98
>  BERT | TRADES Loss | 86.80 | 90.44 | 97.56 | 93.96
> BERT | DR Loss | **89.04** | **93.02**| **97.87** |**94.31**
>
> From Table R1, DR loss can consistently perform better than PGD loss and TRADES loss in four datasets. Especially, DR loss is able to obtain a relatively high increment in terms of certified robust accuracy (CRA) on the datasets with long text samples such as IMDB and SST2. Thus, DR loss is a more suitable choice for certified robust training for its decomposed regularization for each module, which requires each module to be robust under perturbation.
>
> Secondly, we also conduct experiments in the task of defending against adversarial attacks. We employ BERT as the base model and use PGD loss and TRADES loss as baselines. In our experiments, we conduct adversarial attacks by utilizing TextBugger (Li et al., 2019), a commonly used method for conducting text adversarial attacks. Following the existing adversarial attack setup, we randomly draw 1,000 test samples for adversarial example generation. The defense results of different methods are as follows.
>
> Table R2: Comparison of empirical robust accuracy (%) with robust training loss. We also show the corresponding natural accuracy indicated by the parentheses.
>
> Base Model | Loss | IMDB| SST2 | Yelp | AG
> ---|---|---|---|---|---
>  BERT | PGD Loss | 62.0 (87.1) | 87.6 (91.5) | 92.4 (96.9)  | 85.5 (93.7)
>  BERT | TRADES Loss | 57.8 (84.8) | 86.3 (91.7) | 91.9 (97.0) | 84.4 (94.3)
> BERT | DR Loss | **72.7** (86.9)  | **89.8** (93.3) | **96.9** (98.4) | **87.6** (92.9)
>
> From Table R2, the empirical robust accuracy of DR loss is higher than those of PGD loss and TRADES loss. Compared to the baselines in Table R2, the improvement is obtained by the individual robustness supervision for each module provided by DR loss. Such results further show the effectiveness of the designed DR loss in improving model robustness, and we hope they can address the raised concern of the reviewer.
>
> **References**
>
> Madry, A., Makelov, A., Schmidt, L., Tsipras, D., & Vladu, A. (2017). Towards deep learning models resistant to adversarial attacks. arXiv preprint arXiv:1706.06083.
>
> Zhang, H., Yu, Y., Jiao, J., Xing, E., El Ghaoui, L., & Jordan, M. (2019, May). Theoretically principled trade-off between robustness and accuracy. In International conference on machine learning (pp. 7472-7482). PMLR.
>
> Li, J., Ji, S., Du, T., Li, B., & Wang, T. (2019, January). TextBugger: Generating Adversarial Text Against Real-world Applications. In 26th Annual Network and Distributed System Security Symposium.

---

> > ### Comment · Reviewer_FiMX · 2023-08-16
> > **Response to Rebuttal**
> >
> > Thanks for the clarification and updates. I have read other reviews and the authors' rebuttal to them. Given the new information and results in the rebuttal, I'd like raise my rating score to 6.

---

> > > ### Author Response · Authors · 2023-08-16
> > >
> > > Dear Reviewer FiMX,
> > >
> > > We really appreciate your time and effort. We are thankful for your acknowledgment of the effectiveness of our rebuttal response.
> > >
> > > Sincerely,
> > >
> > > Authors

---

### Official Review · Reviewer_ZzE2 · 2023-07-31

**Soundness:** 2 fair
**Presentation:** 2 fair
**Contribution:** 2 fair
**Rating:** 5
**Confidence:** 4

**Summary:**

This paper introduces a unified framework called UniT, aiming to solve the limitations of existing certified robust training pipelines against text adversarial perturbations. The main contribution is that it works in the word embedding space and provides stronger robustness guarantees without extra modules. Additionally, the paper proposes a decoupled regularization (DR) loss to improve the base model's robustness. Experimental results show the effectiveness of the unified framework and DR loss in enhancing certified robust accuracy.

**Strengths:**

This work presents three advantages:

1. It successfully combines the smoothed model in the discrete word space and the latent space, effectively bridging the structural gap between the two spaces and providing a unified approach for certified robustness. It avoids the loose bound problem caused by IBP.

2. The introduction of a decoupled regularization (DR) loss specifically targets improving the robustness of the base model, separating the robustness regularization terms for feature extraction and classifier modules, leading to better performance.

3. Experiments are conducted on used text classification datasets, demonstrating the effectiveness of the proposed unified framework and the DR loss in improving certified robust accuracy.

**Weaknesses:**

See the question part.

**Questions:**

* It appears to be a forced combination of two robust training methods. Can you provide more insight, such as whether our motivation is to unify Type 1 and Type 2 methods, or to address the issue of requiring IBP in Type 2 methods?
* Upon reviewing the supplementary code, it appears that the authors have used transformers for their experiments, specifically focusing on BERT. To provide a more comprehensive evaluation, it would be beneficial to include additional experimental results with other models such as BERT-large, RoBERTa, and ELECTRA. This would further showcase the effectiveness and compatibility of the proposed framework across various state-of-the-art models. (I think it is very easy to switch to other models)
* Can you provide more explanation and justification for why UniT does not require Interval Bound Propagation (IBP), it is essential to understand the core concept of UniT.
* I have also discovered some other methods that were not compared in the paper. (Such as Jiehang Zeng, Xiaoqing Zheng, Jianhan Xu, Linyang Li, Liping Yuan, and Xuanjing Huang. 2021. Certified robustness to text adversarial attacks by randomized [MASK], Yada Pruksachatkun, Satyapriya Krishna, Jwala Dhamala, Rahul Gupta, and Kai-Wei Chang. 2021. Does robustness
improve fairness? Approaching fairness with word substitution robustness methods for text classification. arXiv
preprint arXiv:2106.10826)

**Limitations:**

See above

---

> ### Author Rebuttal · Authors · 2023-08-10
>
> We deeply thank the reviewer for the time spent reading our paper and for recognizing the reasonableness of the proposed unified framework and DR loss. The questions raised in the review are constructive for our paper. We understand that there is some misunderstanding, and we would like to address it and answer each raised question with deadly earnestness and seriousness. We sincerely hope the following response can clarify any misunderstanding and address the posted questions, and we will be more than happy to answer any followed-up questions.
>
> **Q1: Can you provide more insight?**
>
> **A**: With all due respect, we hold that our method is not a forced combination. Our design is based on a detailed analysis of bottlenecks of existing certified robust training by (1) conducting a structural analysis on two types of frameworks to analyze the similarity and difference between them and (2) validating that IBP is the hindrance against obtaining higher certified robust accuracy for Type II training.
>
> We would like to further illustrate our motivation and insights as follows. In this work, our aim is to improve the certified robustness performance of text classifiers, and it depends on two things: (1) the satisfaction of certification condition in randomized mechanisms and (2) the robustness of the base model. As for our motivation and insights:
> - Firstly, after showing the structural analysis in Figure 1, we notice that Type II methods require IBP to conduct the calculation of certification condition, which has a loose bound problem that makes certification condition harder to satisfy, leading to a lower certified robust accuracy.  We thus want to address the IBP issue in Type II methods by removing it from certified robust training and certification. However, removing IBP would mean that a new certification condition is needed, so we further address the certification condition design problem without IBP by using the newly designed one shown in Theorem 1 which can be directly calculated from the embedding space. Thus, one aspect of our motivation is to use the unified viewpoint on certified robust training to address the IBP issue in Type II scenario. And we correspondingly tackle the challenge of designing a certification condition without IBP in Type II scenario based on the independence of each word.
> - Secondly, based on our analysis, both types of certified robust training use CE loss for supervising the robustness training of base model. Nonetheless, CE loss lacks specialized robustness supervision for each module (feature extractor and classifier). Since improving the robustness of the base model also helps improve the certified robustness, we further design a modular regularization term for improving the robustness of each module based on their corresponding responsibility. And the designed DR loss can work well with the designed unified framework in both scenarios.
>
> **Q2:  It would be beneficial to include additional experimental results with other models.**
>
> **A**: We thank the reviewer for the suggestion. Before showing the evaluation suggested by the reviewer, we just want to point out that as mentioned in L160-L163, we focus on using BERT as the base model because it outputs the best performance for existing methods and we follow their setting. As for the additional experimental results with other models, without loss of generality, we take the case of Type I training on IMDB dataset for illustration. The results are shown in Table R1.
>
> Table R1: Comparison of certified robust accuracy (%) with different base models on IMDB.
> |Base Model |SAFER | UniT |
> |---|---|---|
> |BERT-Base  | 85.36 |  89.04|
> |RoBERTa | 89.12 | 89.20|
> |BERT-Large |  87.84 | 89.76|
>
> From Table R1, UniT consistently outperforms baseline SAFER in Type I scenario with different types of base models including the added RoBERTa and BERT-Large, which further validates the effectiveness and compatibility of the proposed framework.
>
> **Q3: Can you provide more explanation and justification for why UniT does not require IBP?**
>
> **A**: UniT does not require IBP because we have proposed a novel certification condition that is calculated directly in the embedding space. Recall that IBP is used for certification in Type II methods such as CISS. Thus, if we have a new way to certify the prediction results without IBP, we do not require IBP in our training and certification. However, designing a certification condition without IBP is challenging. By utilizing the independence of embedding of each word, we propose Eq. (1) to calculate the maximum deviation $\hat{R}$ caused by all synonyms. We correspondingly propose a novel certification condition based on the word embedding space as shown in Theorem 1, which indicates that the prediction result can be certified if the certified radius $R$ calculated from Eq. (3) is greater than $\hat{R}$. Since this certification does not need IBP, UniT does not require IBP.
>
> **Q4: There are some other methods that were not compared in the paper.**
>
> **A**: We thank the reviewer for pointing those papers out, but they are not appropriate for comparison in certified robust accuracy. Firstly, the method proposed by Zeng et al. is to certify the prediction given a limited amount of unconstrained word perturbation, e.g., only 2 words for the SST2 dataset. For our comparison, we concentrate on the scenario where any word can be replaced by its synonyms. The method proposed in that paper cannot handle this situation and is not suitable for being used as the comparison baseline. Secondly, the other mentioned paper by Pruksachatkun et al. is more about a discussion on whether certified robustness helps fairness, and it is not about designing a method for getting certified robust prediction. The proposed method in that paper is supposed to be a baseline for fairness-related methods, which is not related to our task. We will include those papers in the discussion of related work in the final version.

---

> > ### Comment · Reviewer_ZzE2 · 2023-08-15
> >
> > I thank the authors for providing comprehensive and insightful responses. These answers address most of my concerns and emphasize the advantages of the proposed framework. After reading the clarification as well as the other reviews, I decided to raise my initial rating.

---

> > > ### Author Response · Authors · 2023-08-16
> > >
> > > Dear Reviewer ZzE2,
> > >
> > > Thank you for your positive feedback on our rebuttal response! We are very glad that our response has addressed your concerns.
> > >
> > > Sincerely,
> > >
> > > Authors

---

### Author Rebuttal · Authors · 2023-08-10

Dear Reviewers,

We sincerely thank you for your time and effort spent on our paper. We deeply appreciate all your constructive feedback for improving our paper. We have responded to every raised concern with utmost sincerity, deadly seriousness, and the hope that our response can address them. Please find our response to your specific comments and questions in each corresponding separate response. For each response, Q, C, and A are short for question, comment and answer, respectively.

We are excited that most of the reviewers acknowledge the novelty of the designed unified framework UniT and the proposed DR loss for certified robustness training, the effectiveness of our method, and the high quality of our presentation. In our response, we have provided additional illustrations on certain concepts as requested. In addition, we have conducted all requested experiments and put the results and analysis in the corresponding response section. We will accordingly further enhance our paper in the final version with these useful suggestions.

Thanks again for your time and effort!

Best regards,

Authors

---

### Decision · Program_Chairs · 2023-09-21

**Decision:**

Accept (poster)

**Comment:**

The authors propose a robust training framework against text adversarial perturbation. The proposed method makes sense and the results are convincing. All reviewers agree that the paper is above the bar. I vote for acceptance.